# Does workforce explain the relationship between funding and patient experience? A mediation analysis of primary care data in England

Natasha Salant [1], Efthalia Massou [1], Hassan Awan,[2] John Alexander Ford [1]

## ABSTRACT

**Objectives** To determine whether general practitioner (GP) workforce contributes to the link between practice funding and patient experience. Specifically, to determine whether increased practice funding is associated with better patient experience, and to what degree an increase in workforce accounts for this relationship.

**Setting** Primary care practice level analysis of workforce, funding and patient experience of all NHS practices in England.

**Primary and secondary outcome measures** The link between NHS-provided funding to general practice (payments per patient) and patient experience, as per the General Practice Patient Survey, was evaluated. Subsequently, mediation analysis, adjusted for covariates, was used to scrutinise the extent to which GP workforce accounts for this relationship (measured as the number of GPs per 10 000 patients).

**Participants** We included all general practices in England for which there was relevant data for each primary variable. Atypical practices were excluded, such as those with a patient list size of 0 or where the workforce variable was recorded as being more than 3 SD from the mean. After exclusion, 6139 practices were included in the final analysis.

**Results** We found that workforce (GPs per 10 000 population) significantly (p<0.001) acts as a mediator in the effect of practice funding on overall patient experience even after adjusting for rurality, sex and age, and deprivation. On average, the mediated effect constitutes 30% of the total effect of practice funding on patient experience.

**Conclusions** The increase in the number of doctors in primary care in England appears to be a mechanism through which augmented practice funding could positively impact patient experience. Policy initiatives targeting improved patient experience should prioritise considerations related to workforce and practice funding.

¹Public Health and Primary Care, University of Cambridge, Cambridge, UK
²Keele University, Keele, UK

**Correspondence to**
Ms Natasha Salant;
natashasalant@gmail.com

## STRENGTHS AND LIMITATIONS OF THIS STUDY

⇒ This study is the first to explore the association between funding, workforce and patient experience through a mediation analysis.
⇒ Given that publicly available datasets are used for this analysis, the data quality of the analysis suffers the same quality issues as these datasets. For instance, there is known inflation of full-time equivalent figures in the NHS Workforce dataset.
⇒ The funding data is limited to NHS payments to practices and therefore does not capture other payments received by practices, for instance, for training and teaching.

## INTRODUCTION

In recent years, primary care services in England have been subject to increasing demand, tighter budgets and workforce shortages.[1–5] General practitioner (GP) workforce has been found to be associated with better self-reported quality of care[6] and both GP and patient satisfaction.[7] Workforce inequalities have been identified in previous research with the distribution of GPs, paramedics, and other allied health professionals favouring more affluent areas.[7–10]

Patient experience is an important measure of primary health quality because it provides information about the process of care, for instance waiting times and interaction with staff, which cannot be replaced by other indicators.[11] Self-reported continuity of care has been linked to cost-effectiveness, reduced emergency hospital admissions and reduced mortality.[12] Historically, patient satisfaction of general practice in the UK has been high; but it has been in decline since 2015 and the gap in patient satisfaction between socioeconomic groups have been widening.[1 2] On most measures of quality, practices in more deprived areas fare worse: they have lower quality scores, as measured by the Quality Outcomes Framework (QOF), and are more likely to be rated as 'inadequate' by the Care Quality Commission.[1]

It has been suggested that funding inequalities are a key driving factor behind inequalities in workforce and satisfaction.[1 13] Practices

in more deprived areas receive 7% less general practice funding.[1] Capitation funding for general practice is positively associated with overall practice quality and Care Quality Commission ratings.[14] In addition, there is evidence of an association between funding and patient experience, whereby practices that receive less funding have lower levels of satisfaction.[2] However, as it stands, there has been no robust investigation of the mechanisms of this relationship; in other words how lower funding leads to worse patient experience.

Mediation analysis is a statistical method that can be used to evaluate relationships between variables by quantifying the intermediary process through which a predictor variable affects an outcome variable. By using a mediation analysis, we can explore whether the relationship between funding and patient experience could be explained by GP supply. Put simply, we can evaluate whether practices which receive more funding employ more staff, and in turn whether they have better patient reported experience. The impact of socioeconomic deprivation and patient need on the relationships between these variables can also be explored.

## METHODS

### Data sources

Four publicly available datasets were used: 'NHS Payments to General Practice',[15] 'General Practice Workforce',[16] 'GP Patient Survey Data'[17] and 'National General Practice Profiles'.[18]

Funding data consisted of payments from the NHS to individual general practices in England as part of their contract. Total NHS payments made to general practices included the core capitation amount (global sum) and other payment categories including financial incentives (QOF) and payments for premises. The dataset included total payment and payment per registered and weighted patients (calculated using the Carr-Hill formula[19]).

General practice workforce data are uploaded by practices every quarter and the database covers all general practices in England. GP data are available for full time equivalent (FTE) across all GP partners, salaried GPs, training grades and locums.

The GP Patient Survey (GPPS) is a national self-reported patient experience survey of over 900 000 adults per year in England.[20] It is commissioned by NHS England with the aim of monitoring quality of care. The results are weighted by age, sex, region and socioeconomic status to ensure representativeness.[2] Survey results provided practice-level data that is comparable across practices and time.[21]

National General Practice Profiles are a set of approximately 150 general practice level indicators that are produced by the Office for Health Improvement and Disparities for practices across England. The Index of Multiple Deprivation (IMD) score is an area-based measure of socioeconomic deprivation that comprises seven distinct weighted domains: income (22.5%),

employment (22.5%), health deprivation and disability (13.5%), education, skills and training (13.5%), crime (9.3%), barriers to housing and services (9.3%) and living environment (9.3%).[22]

### Variables

#### Predictor

For funding we used payment per registered patient or per weighted patient by dividing 'total NHS payment' by 'registered patient' or 'weighted patient'.

#### Outcome

We used four patient experience variables from the GPPS based on previous studies.[2 13]
1. Proportion of respondents who had a good or very good experience of making an appointment (access).
2. Proportion of respondents who always see their preferred GP (continuity).
3. Proportion of respondents who had trust and confidence in their GP (trust).
4. Proportion of respondents who had an overall good or very good experience (overall experience).

#### Mediator

We used FTE GPs per 10 000 registered or weighted patients as the mediator.

#### Covariates

An adjusted model was undertaken using patient weights and covariates. Weighted payments per patient takes into account patient sex, age, long-standing health condition and rurality, so these were not added as separate variables. IMD was also added as a covariate.

### Sample

As illustrated in online supplemental figure 1, we included all general practices in England with data available. Where practices were entirely missing from payments, workforce and patient experience datasets, they were excluded. In addition, atypical practices were excluded, for example, if the patient list size was 0 or where workforce data were improbable (defined as more the 3 SD from the mean).

### Data merge

Relevant variables from data sources were linked using practice code, and sense-checked by scanning that practice names matched. To account for the lag between funding, employment and patient experience, 2017/2018 funding data were used, December 2018 workforce data and March 2019 patient experience data. Pandemic data were not used because of data quality issues and challenges around accounting for the impact of changes to services.

### Statistical analysis

Three mediation model structures were built using the Baron and Kenny approach[23] also known as the causal steps approach, and all models assumed the causal structure in figures 1 and 2. These were: an unadjusted model

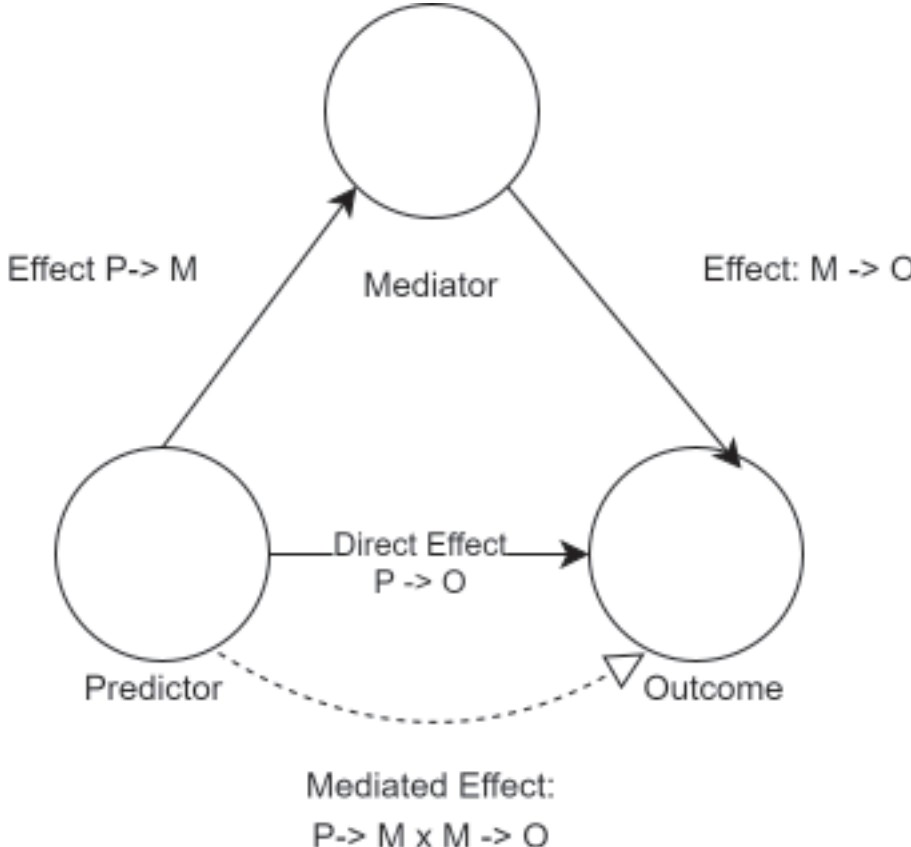

**Figure 1** Assumed causal structure of unadjusted mediation model.

(where GPs per registered patient was tested as a mediator of the effect of payments per registered patient on patient experience), a weighted patients' model (where GPs per weighted patient was tested as a mediator of payments per weighted patient on patient experience) and a deprivation adjusted model (where the unadjusted model was adjusted for deprivation). The weighted model was weighted using the Carr-Hill formula which takes into account, patient sex, age, long-standing health condition and rurality. For each model structure, there were four distinct models that tested each distinct patient experience variable.

In total, 12 mediation models were tested. For each, a simple linear regression of the predictor on the outcome was run to test whether there was a significant association. Mediation analysis was considered if the outcome variable was significantly associated with the predictor variable, the mediator variable was significantly associated with the predictor variable, and controlling for the predictor variable, that the mediator variable associated with the outcome variable. The significance of indirect effect was tested using bootstrapping procedures. Unstandardised indirect effects were computed for each of 1000 bootstrapped samples, and the 95% CI was computed by determining the indirect effects at the 2.5th and 97.5th percentiles. If mediation was established using simple linear regressions, the model was adjusted for patient sex, age, long-standing health condition and rurality.[24 25]

In addition, the model's sensitivity to removing outliers was tested by running the same analyses on datasets that did not remove outliers.

All analyses were conducted using R (V.4.2) and RStudio.

### Patient and public involvement
No patient involved.

### RESULTS
The final number of practices included ranged from 5946 to 6139 (table 1) out of a total of 6836 practices as at 2019. Our results therefore account for approximately 90% of all practices in England. The continuity dataset included the fewest practices because the survey's continuity question was answered by 2% fewer participants than access, trust and overall experience questions. Across all datasets, the average practice receives NHS payments of £152 per registered patient and the average practice size is 8640 patients. Most practices have a General Medical Services contract, do not dispense and are based in urban areas (online supplemental table 1 presents extended practice characteristics). Practices employ on average 5.6 FTE GPs. The average practice receives a good or very good rating of 83% for overall patient experience, 69% for access, 49% for continuity and 95% for trust.

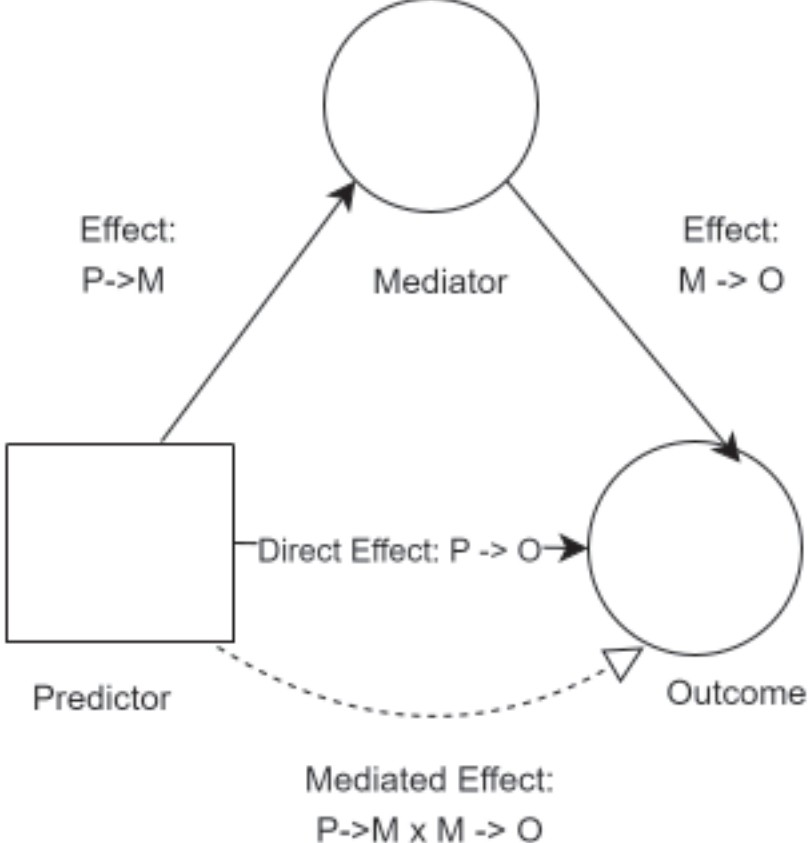

**Figure 2** Assumed causal structure of adjusted mediation model.

### Does workforce mediate the relationship between funding and overall patient outcomes?
#### Unadjusted analysis
In the unadjusted model, the effect of funding on overall patient experience was partially mediated via GPs per 10 000 patients (figure 3). The proportion of the relationship between funding and overall patient experience that is mediated by GPs per 10 000 is 32%. In other words, the increase in payments per patient that drives an increase in GPs per 10 000 patients explains more than one-third of the effect on an increase in overall patient experience.

The regression coefficient between funding and overall patient experience was statistically significant, as was the regression coefficient between GPs per 10 000 patients and overall patient experience (p<0.001). The total effect of funding on overall patient experience was 0.044. The indirect effect via workforce was 0.014 (95% CI 0.011 to 0.017). The model could be seen to illustrate that a £10 increase in practice funding is associated with a 0.44% increase in overall patient experience.

#### Analysis adjusted for socioeconomic deprivation
When adjusted for socioeconomic deprivation, the effect of funding on overall patient experience is still significantly (p<0.001) partially mediated by GPs per 10 000 patients (figure 4). In this adjusted model, the indirect effect via GP workforce was 0.013 (95% CI 0.011 to 0.015). The total effect of funding on overall patient experience (0.043) has the same effect size as the unadjusted model

and is still significant (p<0.001). However, the direct effect (0.030) and indirect effect (0.013), while still both significant (p<0.001), differ slightly in size from the unadjusted model, and the proportion mediated is 30%.

#### Analysis using weighted funding data
Figure 5 shows the effect of funding on overall patient experience was partially mediated via GPs per 10 000 weighted patients. The indirect effect via workforce was 0.010 (95% CI 0.008 to 0.012). This weighted model could be seen to illustrate that a £10 increase in practice funding is associated with a 0.38% increase in overall patient experience, and the proportion of the total effect that weighted funding has on overall patient experience is 26%.

The relationship between weighted funding and overall patient experience (0.028) was significant and similar to the unadjusted model, as was the relationship between GPs per 10 000 patients and overall patient experience (1.152).

### Does workforce mediate the relationship between funding and access, trust and continuity?
#### Unadjusted analysis
Mediation models with access, trust and continuity indicators as outcome variables were all similarly significantly mediated (p<0.001) by GPs per 10 000 patients (online supplemental table 2). The 'access' outcome had the strongest association between funding and patient

**Table 1** Practice characteristics for each outcome dataset

| | Access | Continuity | Trust | Overall experience |
|---|---|---|---|---|
| | Total (N=6137) | Total (N=5946) | Total (N=6137) | Total (N=6139) |
| Payments per patient (£) Mean (SD) | 152 (40.3) | 152 (39.4) | 152 (40.3) | 152 (40.3) |
| Payments per weighted patient (£) Mean (SD) | 151 (35.4) | 151 (34.7) | 151 (35.4) | 151 (35.4) |
| Rurality n (%) | | | | |
| Rural | 919 (15.0%) | 885 (14.9%) | 919 (15.0%) | 919 (15.0%) |
| Urban | 5218 (85.0%) | 5061 (85.1%) | 5218 (85.0%) | 5220 (85.0%) |
| Registered patients Mean (SD) | 8640 (5220) | 8790 (5150) | 8640 (5220) | 8640 (5220) |
| FTE GPs per 10 000 patients (pts) Mean (SD) | 5.61 (2.20) | 5.62 (2.19) | 5.61 (2.20) | 5.61 (2.20) |
| Total patient number >65 years Mean (SD) | 1620 (1210) | 1650 (1200) | 1620 (1210) | 1620 (1210) |
| % patients with long-standing health condition Mean (SD) | 51.3 (8.58) | 51.3 (8.58) | 51.3 (8.58) | 51.3 (8.59) |
| Patient experience rated good (%) Mean (SD) (min, max) | 69.1 (14.4) (19.1, 100) | 49.1 (18.5) (2.14, 98.0) | 95.3 (3.80) (71.6, 100) | 83.4 (9.74) (32.2, 100) |
| Response rate (%) Mean (SD) | 35.9 (10.8) | 36.0 (10.8) | 35.9 (10.8) | 35.8 (10.8) |

FTE, full time equivalent; GP, general practitioner.

experience, with the unadjusted model demonstrating that a £10 pound increase in practice funding was associated with a 0.44% increase in patient experience, of which 36% could be explained by an increase of 1 GP per 10 000 patients.

## Adjusted analysis

When using weighted data and adjusting for deprivation, the 'access' effect size remained the highest out of all models, with GP workforce accounting for ~31% of the effect size. Direct, total and mediated effect sizes were substantially smaller for the continuity and trust models indicating that while an increase in GPs per 10 000 patients was a statistically significant mediator for trust and continuity measures of patient experience, the effect may be more relevant for access and overall experience measures.

Sensitivity analyses including the data outliers did not have a substantial impact on the results.

## Main finding of this study

We found that the number of GPs per 10 000 patients significantly (p<0.001) mediated the effect of practice funding on overall patient experience, access, trust and continuity. In simpler terms, practices with more funding had better patient experience and some of this relationship can be accounted for by an increase in workforce. However, it is crucial to note that the effect sizes are relatively modest. In the unadjusted model, the total effect

size of the association between payments per patient and overall patient experience is 0.044. This implies that a £10 increase in payments per patient is linked to a 0.4% rise in overall patient experience. Considering the mean payment per patient is £152, an average practice would experience only a marginal increase in overall patient experience, even with a funding boost of 6%. Nonetheless, our findings indicate that 30% of the explanation for the link between an increase in payments per patient and improved overall patient experience is attributed to an increase in FTE GPs per 10 000 patients.

## Strengths and limitations

The study was strengthened by building up the mediation model step by step, allowing an understanding of the associations at each level of the mediation. In addition, we used a non-parametric estimation of the indirect effect and significance, which, because of its wide applicability to a variety of models, resolves any doubts arising from whether or not the method of estimation and significance testing is suitable for this model.[24 25]

GPPS response rate is about 30%, raising questions about whether the data is suitable for a quantitative England-wide study. However, because responses are weighted to account for selection bias, demographic characteristics of the eligible population, as well as differences between responders and non-responders,[26] this problem is mitigated. In addition, we have only identified

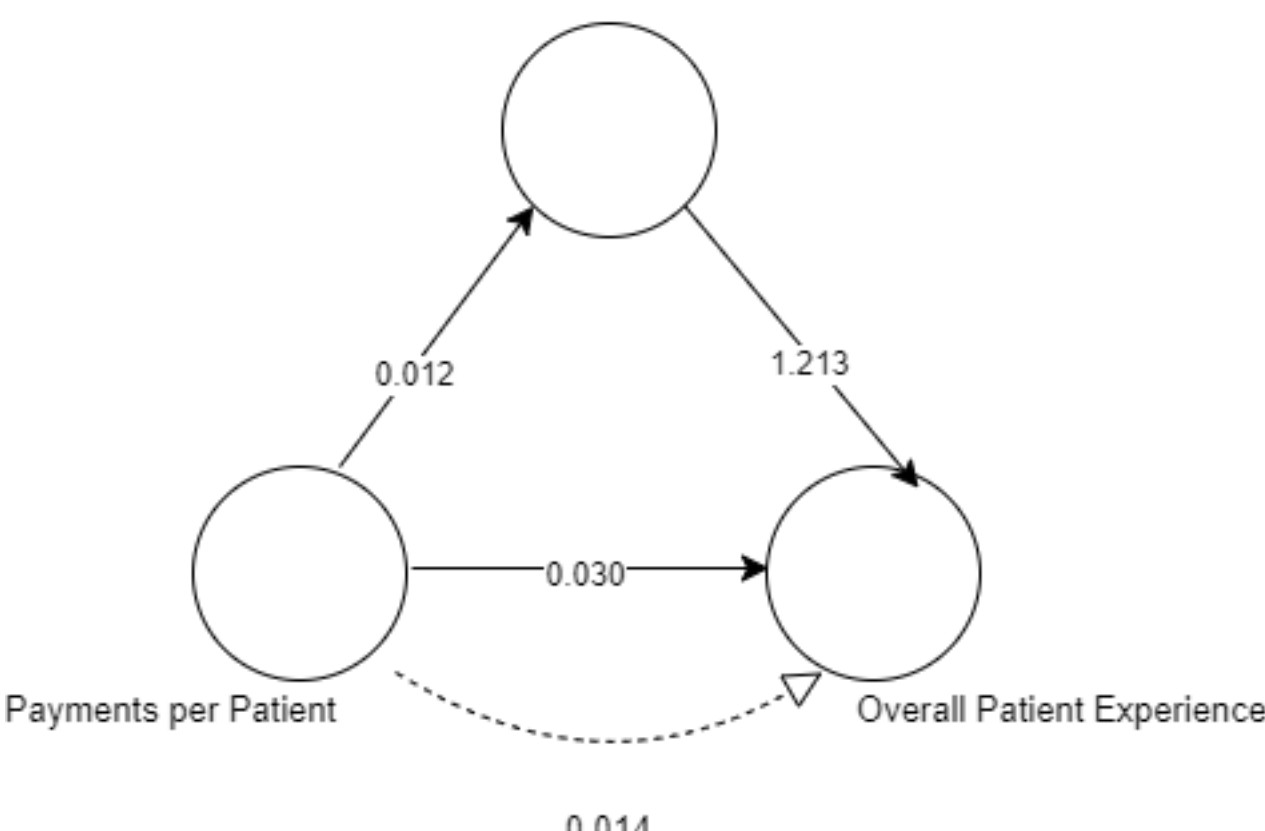

**Figure 3** Unadjusted mediation model results.

four patient experience outcomes for this study, meaning there may be other components which had different results. For example, waiting times may have a stronger relationship with workforce and funding.

Both the General Practice Workforce and NHS Payments datasets have known quality considerations. For the Workforce dataset, there is known inflation of FTE and headcount figures due to the recruitment of temporary staff to cover staff on long-term leave.[27] For the NHS Payments dataset, patient list numbers should be viewed with caution given that there is known list inflation whereby patients who have left a practice are not deregistered.[28]

This analysis of practice funding is limited to an analysis of NHS payments to practice and does not include practice funding received from other sources such as research or training. However, this limitation addresses a wider problem that practice-level data on additional funding, expenditure and profit data is not publicly available.

### What this study adds

This study is the first to explore the association between practice-level NHS payments per patient, patient experience and workforce through mediation analyses. Despite prior evidence showing that an increase in NHS practice funding is associated with an increase in patient experience[2] and CQC scores,[12] it is unclear from these studies how increasing practice funding may lead to improved quality outcomes. Understanding the mechanisms of action is valuable to recognise correlates of increasing GP per patient ratios, identify practices that may be more or less likely to benefit from an increase in payments per patient, or inform policy so that increasing practice funding can be more strategic. Payments per weighted patient are not equal per decile, with more affluent deciles receiving more funding per weighted patient. That information is enriched by the finding that the inequality in payments is associated with an inequality in capability to recruit GPs, which plays a substantial role in patient experience.

Moreover, this study read together with previous analyses on the widening inequality GP distribution[9 10] should be interpreted to mean that if the status quo continues with regard to NHS payments to practices, that is, a total increase in average payments per patient of approximately 4% over the past 4 years,[29] the inequality in payments per weighted patient, as well as that in GPs per 10 000 weighted patients, will likely continue to grow. Given the results of this study, it is unsurprising that recent results from the GPPS reveal a widening gap in patient experience, with patients attending practices in more deprived neighbourhoods reporting worse overall experience than patients attending practices in more affluent neighbourhoods.[30]

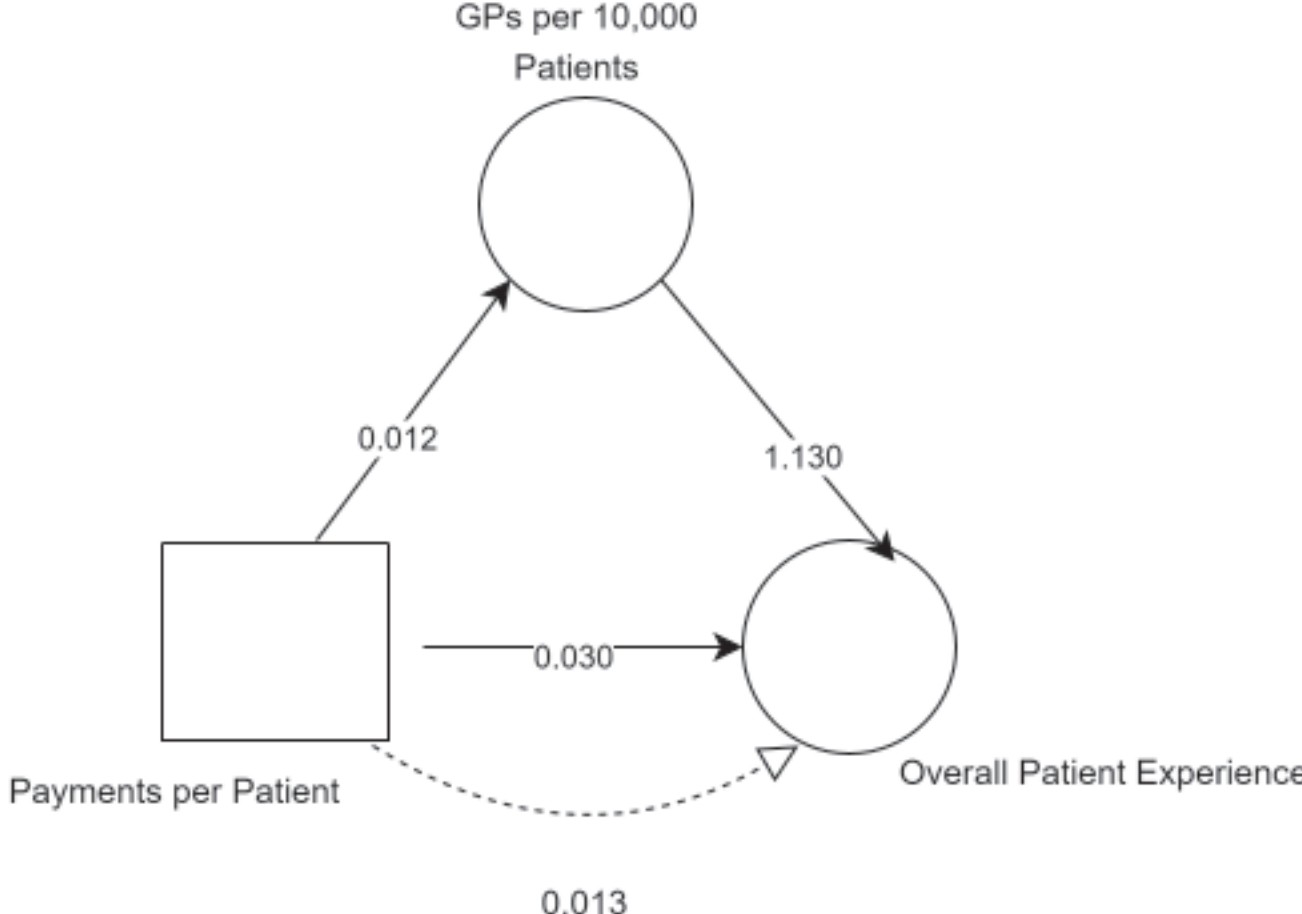

**Figure 4** Deprivation adjusted mediation model results.

Investment in general practice in England is thought to have lagged behind what is necessary to finance increasing total expenditure, population growth, increasing consultations and expanding complexity of health needs.[29] While undeniable value for money can be achieved with primary care investment,[31 32] the results of this study do not suggest that simply investing more money in practices would solve the problem. While practice payments per patient significantly impact the number of GPs per 10 000 patients, the effect size is small. This reflects that workforce recruitment is a problem that is not entirely dependent on practice funding. However, if the system is committed to improving quality, policy must target both practice funding, GP recruitment and quality improvement.

### Policy recommendations
The contractual funding model (Carr-Hill formula) should be reviewed to rectify the unequal distribution of payments per weighted patient. Its inaccurate reflection of patient need becomes apparent in the inequality of average practice payments per weighted patient. As discerned in our analysis this has an impact on the distribution of GPs per 10 000 patients as well as on inequality of patient experience. In addition, given the effect of practice funding on GPs per 10 000 patients

identified by this study, as well as the widening inequality of GP distribution found in multiple previous studies, it is recommended that additional funding is identified and targeted towards underdoctored areas to stop the inequality from increasing. This targeted investment could have a knock-on effect on patient experience in these areas, according to the findings of this study. Finally, given the demonstrable link between funding, GP supply and patient experience, it is recommended that practices achieving low quality of care scores are not penalised by withholding financial incentives. Instead, a supportive approach should be adopted, fostering improvement in the quality of care provided by these practices.

The formulation and implementation of these recommended policy reforms require inclusive input from a diverse cadre of primary care practitioners, as well as the active inclusion of a robust patient perspective.

### CONCLUSION
We found that the number of GPs per 10 000 patients is a significant mediator and explains a third of the relationship between funding and patient experience. However, the effect is small, meaning that a substantial increase in funding would be required to meaningfully improve

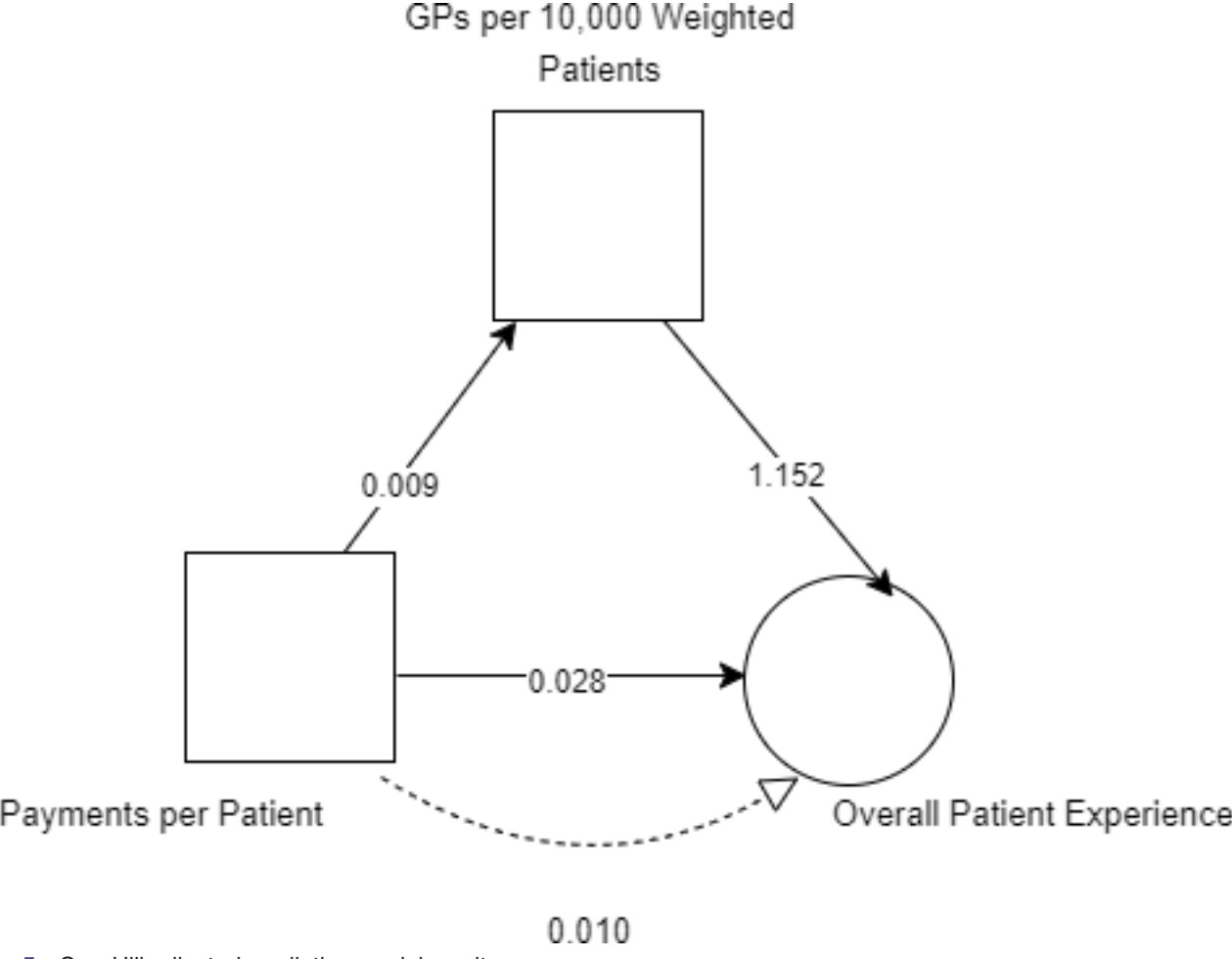

**Figure 5** Carr-Hill adjusted mediation model results.

patient experience. Funding increases should therefore be implemented alongside policy reform to improve patient experience and reduce inequalities.

**Correction notice** This article has been corrected since it was published. Licence updated to CC BY on 2nd August 2024.

**Contributors** NS is the primary author having made substantial contributions to the conception and design of the work, the acquisition of the data, the cleaning and analysis of the data as well as the interpretation thereof, and therefore acts as guarantor. She is also responsible for drafting the work, revising it and final approval for it to be published, thereby agreeing to be accountable for all aspects of the work. HA made substantial contributions to the interpretation of data, the critical revision and final approval for publication thereby agreeing to be accountable for the work. EM made substantial contributions to the design of the statistical analysis as well as the interpretation of the analysis, critical revision of the draft and final approval for publication thereby being accountable for the work. JAF conceptualised the initial idea and made substantial contributions to the concept and design of the work as well as to the interpretation of the data, critical revision of the draft and final approval for publication thereby agreeing to be accountable for the work.

**Funding** NS is grateful to Chevening Fellowships, the UK government's global scholarship programme, funded by the Foreign, Commonwealth and Development Office (FCDO) and partner organisations for the funding of her MPhil programme, thereby making this research possible.

**Competing interests** None declared.

**Patient and public involvement** Patients and/or the public were not involved in the design, or conduct, or reporting, or dissemination plans of this research.

**Patient consent for publication** Not applicable.

**Ethics approval** Not required.

**Provenance and peer review** Not commissioned; externally peer reviewed.

**Data availability statement** Data are available upon reasonable request. Data will be made available upon request. Please contact natashasalant@gmail.com in this regard.

**ORCID iDs**
Natasha Salant http://orcid.org/0000-0002-0475-4278
Efthalia Massou http://orcid.org/0000-0003-0488-482X
John Alexander Ford http://orcid.org/0000-0001-8033-7081

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
