## [Reviewer comments · BMJ Open]

ARTICLE DETAILS

TITLE (PROVISIONAL)	Does workforce explain the relationship between funding and patient experience? A mediation analysis of primary care data in England.
AUTHORS	Salant, Natasha; Massou, Efthalia; Awan, Hassan; Ford, John

VERSION 1 – REVIEW

REVIEWER	Rodríguez , José E University of Utah Health, Family and Preventive Medicine
REVIEW RETURNED	06-Mar-2023

GENERAL COMMENTS	I normally am very supportive of authors in their attempts to get their articles published. However, I found this article difficult to follow and found myself asking “why did they do this work? I will, however, limit my comments to the parts of the article that I could follow. Abstract: This needs to be clearer to the readers. The objective seems to be to look for a relationship between the number of GP’s and patient satisfaction with the NHS. If that is the case, please state. After looking up a mediation analysis, it is clear that the authors are trying to introduce GP’s as a factor that The term “mediation” is confusing here. Does it mean that the GP’s are mediators between the NHS and patients? I almost think that the word that you are looking for is mitigation—do GP’s mitigate decreases in patient satisfaction due to supply shortages? Please clarify. Results: does this mean that even if they are poorly paid, but there are enough of them GP’s make the patient experience better? If so, that is a wonderful endorsement of Family Medicine. My concern is that the writing obfuscates that central idea, if it even is the central idea of the paper. Conclusion: From the abstract it should say something like, if there are more GP’s, even in socially deprived areas, they ensure that patients have greater satisfaction with the NHS. And that is a meaningful conclusion, if that is what your research showed Introduction: This introduction is both lengthy and wordy, and readers will get lost in looking for meaning. Please use the Welch article to frame your introduction. (https://cancer.dartmouth.edu/sites/default/files/2019-05/papertrail.pdf) Answer these three questions in your
--

	introduction: 1. What is the general problem? 2.What is the specific problem? and 3. How does this study help? My read is that there is a problem with patient satisfaction with the NHS (general problem) but I do not know after reading this paper several times how this study helps or what is the specific problem. The last paragraph of the introduction is good and should be kept. Again—not sure mediation is the right word. Methods: Again, the methods section is wordy and complex, as well as difficult to follow. The Welch article is clear on how to write a methods section. The sub-headings help with understanding. I suggest a laymans explanation of mediation analysis, so that people can see that you are using that analysis to show that GP supply is something that increases patient satisfaction. Results: These are well documented in multiple tables, but the text needs to say exactly what was found in clear terms. Again, the welch article will be helpful in simplifying this for a general audience. Conclusion: This study presents a major finding—that GP supply is related to patient satisfaction.
--	---

REVIEWER	Barrera, Lena Universidad del Valle
REVIEW RETURNED	16-Aug-2023

GENERAL COMMENTS	Dear authors Thank you for letting us to review your research. Patient experience have been arousing as an important performance indicator for health services as it is correlated with appropriate health outcomes. Although the researchers made an important effort to understand the routes of patient experience, the following aspects detract from the best inference:  1. It is not clearly the process to select the data to include in the analysis. It is recommendable to include a flow-chart explaining the how the practices were excluded and how was the completeness for each database. 2. It is necessary to mention the concept of patient experience. The author said that the used a questionnaire implemented in the UK but it is not clear what is the theory behind the questionnaire. It could be useful to insert the questionnaire as an appendix. 3. I am not sure if the study measured association, or the results could be factors correlated with patient experience taking into account the variables included in the models. See this reference Altman, N., Krzywinski, M. Association, correlation and causation. Nat Methods 12, 899–900 (2015). 4. It is important to clarify whether or not there is a correlation between disease burden and the level of deprivation 5. Considering the questionnaire used to measure the outcome, it is necessary to acknowledge what components of the patient experience could not be included and mention them in the limitations.
--

REVIEWER	Pannarunothai, Supasit Centre for Health Equity Monitoring Foundation22-Oct-2023
REVIEW RETURNED	22-Oct-2023

GENERAL COMMENTS	The paper specifically asks the mediation effects of the number of general practitioners per 10000 patients on patient’s experiences with primary care services. With the available primary care administrative databases, the research linked datasets at the primary care practice level to answer research questions. Non-parametric bootstrapping regression analyses provide first evidence on the mediation effects of number of workforces per 10000 patients as expected. Policy recommendations are related to research findings. Minor revision is required to make readers satisfied with the quality of the presentation. Strengths and limitations Page 3, lines 52-53: PCN and CCG need explanation. Introduction Page 5, lines 7 and 11: references should start with 1, 2 rather than 35, 36. Methods Page 6, line 15: the authors should clarify the term “patients” that used to reflect the populations registered to general practice. Because all people who choose the primary care practice may not necessarily visit their GP over the observed period. Results Page 9, lines 37-39: all acronyms need explanation. Line 55: it is doubtful why the number in column continuity was higher that other columns while the authors explained in text that the question on continuity was 2% responded by the samples. Figure and legend Page 15, lines 22 and 24: figures 4 and 5 are in the wrong order
---

VERSION 1 – AUTHOR RESPONSE

Reviewer 1	Abstract: This needs to be clearer to the readers. The objective seems to be to look for a relationship between the number of GP’s and patient satisfaction with the NHS. If that is the case, please state.	We agree and have substantially revised the abstract to improve clarity including specifying that the objective is to determine whether increased practice funding is associated with better patient experience and to what extent an increase in workforce explains this relationship .
	After looking up a mediation analysis, it is clear that the authors are trying to introduce GP’s as a factor that The term “mediation” is confusing here. Does it	Thanks for highlighting this. By mediation we meant a statistical tool that can be used

	mean that the GP's are mediators between the NHS and patients? I almost think that the word that you are looking for is mitigation—do GP's mitigate decreases in patient satisfaction due to supply shortages? Please clarify.	to evaluate relationships between variables by quantifying the mechanism through which a predictor variable affects an outcome variable. We have highlighted this in the text.
	Results: does this mean that even if they are poorly paid, but there are enough of them GP's make the patient experience better? If so, that is a wonderful endorsement of Family Medicine. My concern is that the writing obfuscates that central idea, if it even is the central idea of the paper.	We did not include data on payment of GPs as this data is not publicly available. We included data on funding of general practice as independent organisations commissioned by the NHS to provide primary care services. Undoubtedly funding to practices is related to payment of GPs. We have substantially re-drafted the manuscript to make the methods and results clearer.
	Conclusion: From the abstract it should say something like, if there are more GP's, even in socially deprived areas, they ensure that patients have greater satisfaction with the NHS. And that is a meaningful conclusion, if that is what your research showed	We have reworded the conclusion accordingly because we agree that the research is meaningful and contributes to a growing body of evidence. Our conclusion is that an increase in the number of doctors in primary care in England may be a mechanism through which increased practice funding could affect increased patient experience in England. Policy efforts to address patient experience should focus on workforce and practice funding.
	This introduction is both lengthy and wordy, and readers will get lost in looking for meaning. Please use the Welch article to frame your introduction. (https://cancer.dartmouth.edu/sites/default/files/2019-05/papertrail.pdf) Answer these three questions in your introduction: 1. What is the general problem? 2. What is the specific problem? and 3. How does this study help? My read is that there is a problem with patient satisfaction with the NHS (general problem) but I do not know after reading this paper several times how this study helps or what is the specific problem. The last paragraph of the introduction is good and should be kept. Again—not sure mediation is the right word.	We have edited the whole introduction in order to improve flow, understanding and bring out the key questions you've suggested we focus on. We have also added a paragraph which should help with meaning with regard to the mediation analysis as mentioned above.
	Again, the methods section is wordy and complex, as well as difficult to follow. The Welch article is	We have revised the methods section to improve the clarity.

	clear on how to write a methods section. The sub-headings help with understanding. I suggest a laymans explanation of mediation analysis, so that people can see that you are using that analysis to show that GP supply is something that increases patient satisfaction.	Some of the technical statistical detail is necessary because there are different approaches to mediation analyses, but we have tried to clarify meaning in simple terms as far as possible.
	These are well documented in multiple tables, but the text needs to say exactly what was found in clear terms. Again, the welch article will be helpful in simplifying this for a general audience.	We've simplified the tables and moved additional material to a supplementary appendix.
	This study presents a major finding—that GP supply is related to patient satisfaction.	We agree that this is one major finding. In addition we have found that GP supply (or workforce) statistically significantly explains the relationship between funding and patient experience.
Reviewer 2	1. It is not clearly the process to select the data to include in the analysis. It is recommendable to include a flow-chart explaining the how the practices were excluded and how was the completeness for each database.	We have inserted a flow chart to make this process clearer.
	2. It is necessary to mention the concept of patient experience. The author said that the used a questionnaire implemented in the UK but it is not clear what is the theory behind the questionnaire. It could be useful to insert the questionnaire as an appendix.	We understand that a reader who may have had no exposure to the survey may want to understand more about it. Here is a link to the survey: https://gp-patient.co.uk/surveysandreports . For the purposes of this research, we think that the description of the outcome variables might be enough to explain what elements of patient satisfaction were being explored here. The questionnaire changes slightly every year (though not the particular questions we have referred to) and therefore we think it would be of less value adding it as an appendix
	3. I am not sure if the study measured association, or the results could be factors correlated with patient experience taking into account the variables included in the models. See this reference Altman, N., Krzywinski, M. Association, correlation and causation. Nat Methods 12, 899–900 (2015).	The analysis we conducted, a mediation analysis, measures the association between variables using regression, however the difference is that it combines regression coefficients to examine hypothesised causal pathways. In our study our hypothesized

		casual pathway is that increased funding leads to more workforce and better outcomes. Therefore while it does report association between variables, it does so along a causal pathway and so we are able to make stronger inferences about possible causal relationships.
	4. It is important to clarify whether or not there is a correlation between disease burden and the level of deprivation	We have added some evidence to point to the established correlation between disease burden and deprivation, but we have not ourselves analysed this as it is out of scope of this study.
	5. Considering the questionnaire used to measure the outcome, it is necessary to acknowledge what components of the patient experience could not be included and mention them in the limitations.	We have added a sentence to the limitations to say that we only looked at four patient experience outcomes, and that there may be other components which had different results. For example, waiting times may have a stronger relationship with workforce and funding. We have also clearly stipulated in the methods section which outcomes we are considering.
Reviewer 3	Strengths and limitations: Page 3, lines 52-53: PCN and CCG need explanation.	We have deleted this as we felt it was unnecessary information.
	Introduction: Page 5, lines 7 and 11: references should start with 1, 2 rather than 35, 36.	We have addressed this problem.
	Methods: Page 6, line 15: the authors should clarify the term "patients" that used to reflect the populations registered to general practice. Because all people who choose the primary care practice may not necessarily visit their GP over the observed period.	We use the term patients because it reflects the language of all the datasets we use (for instance weighted patients, and payments per patient). The funding and workforce data includes those who have accessed or not, but only includes those registered. We haven't included unregistered patients in our analysis, however this is likely to be a small number in the UK because primary care is free at the point of care for the majority of the population.
	Results: Page 9, lines 37-39: all acronyms need explanation.	We have now removed information that is not essential to the primary analysis from the

		manuscript and will insert it as an appendix in case some readers who are exposed to these variables (like primary care contract types) may wish to know this information.
	Results: Line 55: it is doubtful why the number in column continuity was higher than other columns while the authors explained in text that the question on continuity was 2% responded by the samples.	Having removed unnecessary information from the table we hope that this aids understanding. We have also explained in text that the continuity column has fewer samples because it has fewer responses.
	Page 15, lines 22 and 24: figures 4 and 5 are in the wrong order	We have addressed this.

VERSION 2 – REVIEW

REVIEWER	Pannarunothai, Supasit Centre for Health Equity Monitoring Foundation
REVIEW RETURNED	05-Dec-2023

GENERAL COMMENTS	The paper has been revised according to previous comments. A few details if added would help clearer understandings. The selected 6,135 general practices were accounted for how many percentage of the total practices in England. The authors propose in the policy recommendations that the Carr-Hill formula should be reviewed. As the resource allocation formulae have been implemented very long period since 1976 and followed by Carr-Hill formula, what mechanisms have been in place in the review processes that number of GPs and user voices would be take place in the future formulae.
---

VERSION 2 – AUTHOR RESPONSE

Reviewer	Comment	Response
Reviewer 3	The paper has been revised according to previous comments. A few details if added would help clearer understandings. The selected 6,135 general practices were accounted for how many percentage of the total practices in England.	Thank you for this suggestion. We have now added some extra words into the results section to put the number of practices into context. 6139 practices constitutes approximately 90% of total practices in England as at 2019.

	The authors propose in the policy recommendations that the Carr-Hill formula should be reviewed. As the resource allocation formulae have been implemented very long period since 1976 and followed by Carr-Hill formula, what mechanisms have been in place in the review processes that number of GPs and user voices would be take place in the future formulae.	We agree that any formula reform would need broad and representative GP input and a strong patient voice and as such have included this in the policy recommendation paragraph.
--	--	--